# Early Extra-Uterine Growth Restriction in Very-Low-Birth-Weight Neonates with Normal or Mildly Abnormal Brain MRI: Effects on a 2–3-Year Neurodevelopmental Outcome

**DOI:** 10.3390/nu16030449

**Published:** 2024-02-03

**Authors:** Paolo Massirio, Marcella Battaglini, Irene Bonato, Sara De Crescenzo, Maria Grazia Calevo, Mariya Malova, Samuele Caruggi, Alessandro Parodi, Deborah Preiti, Agata Zoia, Sara Uccella, Domenico Tortora, Mariasavina Severino, Andrea Rossi, Cristina Traggiai, Lino Nobili, Pasquale Striano, Luca Antonio Ramenghi

**Affiliations:** 1Neonatal Intensive Care Unit, Maternal and Neonatal Department, IRCCS Istituto Giannina Gaslini, 16147 Genoa, Italy; marcellabattaglini@gaslini.org (M.B.); irenebonato@gaslini.org (I.B.); sara.decrescenzo2@studio.unibo.it (S.D.C.); samuelecaruggi@gaslini.org (S.C.); alessandroparodi@gaslini.org (A.P.); zoiaagata@gmail.com (A.Z.); lucaramenghi@gaslini.org (L.A.R.); 2Department of Neurosciences, Rehabilitation, Ophthalmology, Genetics, Maternal and Child Health (DINOGMI), University of Genoa, 16132 Genoa, Italy; sarauccella@gaslini.org (S.U.); linonobili@gaslini.org (L.N.); pasqualestriano@gaslini.org (P.S.); 3Epidemiology and Biostatistic Unit, Scientific Direction, IRCCS Istituto Giannina Gaslini, 16147 Genoa, Italy; mariagraziacalevo@gaslini.org; 4Psychology Unit, IRCCS Istituto Giannina Gaslini, 16147 Genoa, Italy; deborah.preiti@gmail.com; 5Child Neuropsychiatry Unit, IRCCS Istituto Giannina Gaslini, 16147 Genoa, Italy; 6Neuroradiology Unit, IRCCS Istituto Giannina Gaslini, 16147 Genoa, Italy; domenicotortora@gaslini.org (D.T.); mariasavinaseverino@gaslini.org (M.S.); andrearossi@gaslini.org (A.R.); 7Department of Health Sciences (DISSAL), University of Genoa, 16132 Genoa, Italy; 8Neonatology Unit, International Evangelical Hospital, 16122 Genoa, Italy; cristinatraggiai@gaslini.org; 9Paediatric Neurology and Muscle Disease Unit, IRCCS Istituto Giannina Gaslini, 16147 Genoa, Italy

**Keywords:** preterm, extra-uterine growth restriction, EUGR, VLBW, neurodevelopment, weaning, nutritional education

## Abstract

Extra-uterine growth restriction (EUGR) is a common complication and a known risk factor for impaired development in very-low-birth-weight (VLBW) neonates. We report a population of 288 patients with no or with low-grade MRI lesions scanned at a term equivalent age (TEA) born between 2012 and 2018. Griffiths Mental Development Scale II (GMDS II) at 2 and 3 years, preterm complications and weight growth were retrospectively analyzed. EUGR was defined for weight z-score ˂ 10 percentile at TEA, 6 and 12 months of correct age or as z-score decreased by 1-point standard deviation (SDS) from birth to TEA and from TEA to 6 months. Multivariate analysis showed that a higher weight z-score at 6 months is protective for the global developmental quotient (DQ) at 2 years (OR 0.74; CI 95% 0.59–0.93; *p* = 0.01). EUGR at 6 months was associated with worse locomotor, personal/social, language and performance DQ at 2 years and worse language and practical reasoning DQ at 3 years. In conclusion, a worse weight z-score at 6 months of age seems to be an independent risk factor for significantly reduced GMDS in many areas. These results suggest that we should invest more into post-discharge nutrition, optimizing family nutritional education.

## 1. Introduction

The survival rate in preterm babies, particularly for very-low-birth-weight (VLBW, with birth weight < 1500 g) preterm infants, has drastically increased over the years [1,2,3]. However, the incidence of neurological sequelae like cerebral palsy, cognitive impairment and developmental co-ordination disorder has remained high due to increased survival at a lower gestational age [4,5,6]. Neurodevelopment can be affected by several neonatal complications in preterm babies.

Extra-uterine growth restriction (EUGR) is a common complication in preterm infants [7] and is commonly considered a risk factor for poor development. It is defined as “cross-sectional” when the patient weights below a specific cut-off at a specific time point [8] or “longitudinal” when there is a growth deficit from birth concerning another defined time point [9,10]. Its incidence varies from 13% to 97% in different populations and according to relative definitions [11]. In addition, Ehrenkranz RA et al. observed an increase in the incidence of cerebral palsy, abnormal mental development index and psychomotor development index, as well as neurodevelopment impairment, in patients with EUGR during NICU hospitalization in a large population of ELBW infants [12]. Furthermore, Guellec I et al. demonstrated an increased risk of cerebral palsy in “longitudinal” EUGR between birth and 6 months in a large population of neonates from the EPIPAGE study cohort [13]. Therefore, despite the influence of EUGR in neurodevelopment being widely shared, there is no univocal agreement on which its definition and time point better predict the neurological outcome [10]. Furthermore, the most important studies do not consider MRIs, which is the gold standard to define a diagnosis, including minor brain lesions often missed at ultrasounds [14,15].

The nutritional strategy adopted in the NICU can play an important role not only in the prevention of EUGR but also in the neurological development of patients with VLBW. In particular, the use of fortified breast milk or, as a second choice, enriched formulas for premature infants for enteral nutrition and the early start of parenteral nutrition appear to have a positive impact on the neonates’ neurodevelopment. However, there is lack of evidence regarding the role of post-discharge nutrition [16].

Major brain lesions like cystic periventricular leukomalacia (PVL), germinal matrix-intraventricular hemorrhage (GMH-IVH) complicated by periventricular hemorrhagic venous infraction (PVHI), or post-hemorrhagic ventricular distension (PHVD) and massive cerebellar hemorrhage (CBH) are strongly associated with important neurological impairment [17,18,19,20,21,22,23]. The roles of minor lesions like low-grade GMH-IVH, punctate white matter lesions (PWMLs) and micro CBH are still debated, although some recent works seems to be associated with an impaired neurological development [24,25].

Furthermore, other neonatal complications can play a role in developing neurological sequelae in preterm babies, particularly hypoxia at birth [26], neonatal sepsis [27], necrotizing enterocolitis (NEC) [28,29], bronchopulmonary dysplasia [30,31,32] and surgical procedures [33,34].

The primary aim of our study is to define if EUGR can be an independent prognostic factor for a neurological outcome in preterm neonates with VLBW with negative or minor brain lesions at MRI. Furthermore, the secondary aim of this study is to identify if there is a more sensitive time point of EUGR diagnosis that can predict neurological outcomes more optimally.

## 2. Materials and Methods

### 2.1. Population and Data Collection

All infants with VLBW consecutively admitted to Neonatal Intensive Care Unit of IRCSS Giannina Gaslini Institute who underwent routine brain MRI at term-equivalent age from January 2012 to December 2018 were selected for this study. As per standard internal protocol, MRI scans were performed at term-equivalent age (TEA, between 38 and 42 weeks post-menstrual age) as a part of the screening program for identification of prematurity-related lesions. “Feed and wrap” technique was used to perform the MRI [35]. The need for sedation (oral midazolam, 0.1 mg/kg) to prevent head motion was agreed with by the neuro-radiologist case by case. Scans were performed with a 1.5 Tesla MR system (InteraAchieva 2.6; Philips, Best, The Netherlands) using a dedicated pediatric head/spine coil. Our institutional standard MRI protocol included 3 mm thick axial T2-weighted and T1-weighted images, coronal T2-weighted images, sagittal T1-weighted images, axial diffusion-weighted images (*b* value:1000 s/mm^2^) and axial SWI (susceptibility-weighted imaging), which is the gold standard to identify hemosiderin and low-grade hemorrhage [14]. Patients with major brain lesions such as periventricular leukomalacia (PVL), periventricular hemorrhagic infarction (PVHI), post-hemorrhagic ventricular distention (PHVD) and massive-limited CBH [23]—known to significantly affect neurological outcome—or with congenital brain malformations were excluded from this study. Patients with minor brain lesions such as low-grade GMH-IVH (I-II grade for Volpe’s Classification [20]), punctate white matter lesions (PWMLs) [24] or micro-CBH were included [21].

Demographic and clinical data of the enrolled patients were extracted from clinical charts. Collected data included birth weight, gestational age, sex, mode of delivery, Apgar at 5th minute, diagnosis of neonatal complications (sepsis, NEC and BPD), need of major surgery and breast milk feeding at TEA. Neonatal sepsis was defined by the need for antibiotic therapy for clinical and laboratory findings suggesting blood infection. NEC was defined by clinical and imaging signs suggesting enterocolitis. BPD was defined by the need of any ventilatory support or O_2_ supplementation at 36 weeks. MRI data were collected, and patients were divided into two groups comprising patients with normal MRI and patients with minor brain lesions, like the very mild hemorrhages identified with susceptibility-weighted imaging [14,15].

Anthropometric data were collected by clinical charts of patients enrolled in our preterm follow-up service. During the hospital stay, weight was measured daily; after discharge, it was measured at term-equivalent age (TEA) and 1, 3, 6, 9 and 12 months of correct age. We collected only weight at birth, at TEA, and 6 and 12 months of corrected age for the statistical analysis. Z-scores of weights for age and sex were calculated for birth and TEA using INTERGROWTH-21 relatives charts for very low weight at birth [36] and postnatal growth standard in preterm infants [37], while for 6 and 12 months of correct age, we used CDC Growth Charts, 2000 [38]. Patients who were lost in the follow-up phase or with incomplete anthropometric data were excluded from this study.

The patients with birth weight z-score < 1.282 (<10° percentile) were considered small for gestational age (SGA). Extra-uterine growth restriction (EUGR) at TEA and 6 months was diagnosed by two different definitions: “cross-sectional” EUGR as a weight z-score < 1.282 (<10° percentile) and “longitudinal” EUGR as z-score decreased by 1-point SDS from birth to TEA and from TEA to 6 months [39]. A weight with z-score < 1.282 (<10° percentile) was also defined as EUGR at 12 months of correct age.

The neurological development evaluation of patients was performed with Griffiths Mental Developmental Scale II (GMDS II) [40] at two years of corrected age and three years of chronological age. Patients lost in the follow-up phase were excluded from this study. Patients who did not perform the 3 years assessment were included. These evaluations are part of routine follow-up service offered to all patients with VLBW after discharge from the hospital. GMDS was administered by a 10-year experienced single operator blinded to MRI results. Raw numbers were converted into standardized development quotients (DQs). Total development quotients (DQs), relating to global development, were derived from the mean of the results of different areas of assessment: locomotor and gross motor skills (scale A); personal/social and adaptive behavior development (scale B); receptive/expressive language (scale C); fine motor function and hand–eye coordination (scale D); performance as precursors of reasoning and planning (scale E); and practical reasoning (scale F, performed only at 3 years of age). Resulting values were used to evaluate the level of neurodevelopment: values below 70 define a developmental delay, values from 70 to 84 evidence a borderline condition and values above 84 are considered normal [40].

### 2.2. Statistical Analysis

Descriptive statistics were generated for the whole cohort; data were expressed as the mean and standard deviation for continuous variables and absolute and relative frequencies for categorical variables. All collected demographic and clinical data were compared using the chi-squared test or Fisher’s exact test and the Mann–Whitney U test for categorical and continuous variables. Univariate analysis determined the potential risk factors which were significantly associated with unsatisfactory scores in the GMDS (<85) at 2 and 3 years of CA. Logistic regression analysis was used for each variable, and odds ratios (ORs) were calculated with 95% confidence intervals (CIs). The absence of exposure to the factor or the variable that was less likely to be associated with the risk was used as a reference for each analysis. Multivariate analysis corrected for gestational age (GA) was then performed. The only variables that proved to be statistically or borderline significant in the univariate analysis (<0.08) were included in the model. The best-fit model was based on the backward stepwise selection procedures, and each variable was removed if it did not contribute significantly. In the final model, a *p*-value of <0.05 was considered statistically significant, and all *p*-values were based on two-tailed tests. Statistical analysis was performed using the Statistical Package for the Social Sciences for Windows (SPSS 29 - 2022, Sept. Inc., Chicago, IL, USA).

The studies involving human participants were reviewed and approved by Giannina Gaslini Hospital, Genoa, Italy. Written informed consent to participate in this study was provided by the participants’ legal guardian/next of kin.

## 3. Results

### 3.1. Main Population Description

In total, 498 very-low-birth-weight neonates underwent brain MRIs from January 2012 to December 2018 (mean GA 28.7 ± 2.3; mean birth weight 1081 g ± 265 g). Of these patients, 65 were excluded for severe brain lesions, 18 for incidental findings of brain malformations and congenital diseases, and 127 for incomplete follow-up or missing data.

The final population included 288 patients. Mean GA was 28.9 ± 2.1 weeks (range 23–34.6), with a mean birth weight of 1097 ± 255 g (z-score −0.449 ± 1.09; range 435–1490 g). Of these, 139 patients (48.3%) were male. The incidence of SGA was 21.5% (62 patients). Moreover, 232 neonates were born via a cesarean delivery (80.6%), and mean Apgar score at 5 min was 8 (range 2–10). Regarding the major neonatal complications of preterm, incidence of sepsis was 37.5% (*n* = 108), and NEC was present in 30 patients (10.4%), with 15 of these having to undergo surgical treatment. A total of 36 patients (12.5%) underwent surgery before discharge (15 for NEC, 17 for patent ductus arteriosus, PDA). The MRI study at term age (TEA) showed that 101 patients had low-grade lesions (35.1%). Of these, 44 patients had low-grade IVH (15.3%), 47 had punctate lesions of white matter (16.3%) and 31 had micro-CBH (10.8%). At term age, only 17% (*n* = 49) were fed by breast milk exclusively, and 41% were fed only with formula milk. The mean weight at TEA was 2600 ± 598 g (range 1140–4180 g), the mean z-score for weight was −1.407 ± 1.415 and 50% (*n* = 144) had a “cross-sectional” EUGR. The incidence of “longitudinal” birth to TEA was 43.8% (*n* = 126). At 6 months of age, the mean weight was 6.81 ± 1.03 kg, z-score −1.24 ± 1.29 and the incidence of patients with a “cross-sectional” EUGR was 46.2% (*n* = 133), while the rate of the “longitudinal” EUGR from TEA to 6 months was 16.0% (*n* = 46). At 12 months of age, the mean weight was 8.78 ± 1.16 kg, z-score −1.36 ± 1.25, and the incidence of patients with a “cross-sectional” EUGR was 48.3% (*n* = 139) (Table 1).

### 3.2. Neurological Outcome and Statistical Analysis

In relation to the long-term neurological outcomes, the global DQ evaluated with Griffiths Scale II was <85 in 56 patients of the total 288 at 2 years of age (19.4%). Considering the different rating areas of the Griffiths Scale II separately, the incidence of development delay or borderline (DQ < 85) was 23.6% for locomotor, 23.2% for personal/social; 44.1% for hearing and language, 13.2% for hand–eye coordination and 24.3% for performance. At 3 years of age, the global DQ Griffiths scale was <85 in 100 of the 262 patients (26 patients were lost in the follow-up phase) with an incidence of 38.2%. Considering the different rating areas separately, the incidence of development delay or borderline (DQ < 85) was 31.7% for locomotor, 33.2% for personal/social; 53.0% for hearing and language, 31.6% for hand–eye coordination, 48.1% for performance and 38.5% for practical reasoning (Table 2).

Considering the risk factors for worse neurological outcomes at 2 years, the univariate analysis showed that patients scoring below 85 on the Griffiths Scale II had a lower weight z-score at 6 (−1.639 vs. 1.140 *p* = 0.03) and 12 months (−1.735 vs. −1.264 *p* = 0.03) than patients with normal results on the Griffiths assessment, and the incidence of patients with a “cross-sectional” EUGR at 6 months was higher and not statistically significant in patients with developmental delay or borderline global DQ on the Griffiths scale (58.9 vs. 43.5%; *p* = 0.07). Furthermore, the incidence of surgical NEC appears to be higher in patients with development delay or borderline global DQ (10.7% vs. 3.9%; *p* = 0.08). Based on these data, multivariate analysis adjusted for gestational age showed that a higher z-score for weight at 6 months would be protective for a global DQ deficit at 2 years of age (OR 0.74; CI 0.59–0.93; *p* = 0.01) (Table 3).

The same analysis was performed separately for the different rating areas of the Griffiths Scale II. The results of the multivariate analysis are resumed in Table 4. Considering the locomotor area (scale A), the multivariate analysis showed that the “cross-sectional” EUGR at 6 months (OR 1.96; CI 95% 1.10–3.47; *p* = 0.02), punctate white matter lesions (PWMLs) (OR 2.33; CI 95% 1.15–4.71; *p* = 0.02) and major surgery during NICU stay (OR 3.79; CI 95% 1.69–8.49; *p* = 0.001) were a negative prognostic factor. Considering the personal/social area (scale B), the “cross-sectional” EUGR at 6 months (OR 1.94; CI 95%; 1.12–3.37; *p* = 0.02) and NEC (OR 2.6; CI 95% 1.14–5.92; *p* = 0.02) are the major risk factors. Considering the hearing and language areas (scale C), the multivariate analysis identified higher birth weight z-scores (OR 0.31; CI 95% 0.12–0.81; *p* = 0.02) as a protective factor, while the presence of NEC (OR 2.48; CI 95% 1.07–5.71; *p* = 0.03) and “cross-sectional” EUGR at 6 months (OR 1.87; CI 1.05–3.29; *p* = 0.02) were also confirmed as negative prognostic factors. Considering hand–eye coordination assessment (scale D), NEC was also a negative prognostic factor (OR 3.98; CI 1.66–9.55; *p* = 0.002). Lastly, considering performance area (scale E), male sex (OR 2.01; CI 95% 1.13–3.57; *p* = 0.02), major surgery (OR 4.07: CI 95% 1.78–9.33; *p* = 0.001), punctate white matter lesions at MRI (OR 2.03; CI 95% 1.00–4.14; *p* = 0.05) and “longitudinal” EUGR at 6 months (OR 2.10; CI 95% 1.03–4.30; *p* = 0.04) seemed to be negative prognostic factors (Table 4).

Regarding the neurological outcome at 3 years of age, the Griffiths Scale II was applied only in 262 of the 288 patients because 26 patients were lost in the follow-up phase. The univariate analysis showed a major incidence in the Griffiths score of <85 in the male babies (60% vs. 43.2%; *p* = 0.01) and patients with NEC (17% vs. 7.4%; *p* = 0.02), while normal Griffiths had a higher incidence in patients born via cesarean delivery (76% vs. 85.2%; *p* = 0.05). Based on these data, the multivariate analysis adjusted for gestational age showed that only male sex (OR 1.94; CI 95% 1.16–3.24; *p* = 0.01) and NEC (OR 2.55; CI 95% 1.11–5.86; *p* = 0.03) were independent negative prognostic factors for worse global DQ at 3 years (Table 5).

Considering the areas of the Griffith Scale II at 3 years separately, the multivariate analysis showed that, considering locomotor area (scale A), male sex was a negative prognostic factor (OR 1.82; CI 95% 1.07–3.10; *p* = 0.03). Considering the personal social area (scale B), male sex was a negative prognostic factor (OR 2.18; CI 95% 1.28–3.72; *p* = 0.004) while cesarean delivery seemed to be a protective factor (OR 0.47; CI 95% 0.25–0.91; *p* = 0.02). Considering hearing and language area (scale C), “cross-sectional” EUGR at 6 months (OR 1.63; CI 95% 0.99–2.68; *p* = 0.05) and male sex (OR 1.88; CI 95% 1.14–3.10; *p* = 0.01) seemed to be the only negative prognostic factors. Considering hand–eye coordination (scale D) and performance (scale E), the major risk factors seemed to be male sex (OR 4.17; CI 95% 1.78–9.76; *p* = 0.001—OR 2.39; CI 95% 1.44–3.97; *p* = 0.001, respectively and NEC (OR 4.17; CI 95% 1.78–9.76; *p* = 0.001—OR 4.31; CI 95% 1.63–11.35; *p* = 0.003, respectively). Considering practical reasoning (scale F), only “longitudinal” EUGR at 6 months (OR 2.07; CI 95% 1.02–4.17; *p* = 0.04) and NEC (OR 4.47; CI 95% 1.84–10.85; *p* = 0.001) seemed to be significant (Table 6).

## 4. Discussion

EUGR is already considered a risk factor for worse global mental development [7,8,9,10,11,12]. In our population, we found a high incidence of EUGR, particularly at TEA (Table 1). Despite this, EUGR at TEA (both “cross-sectional” and “longitudinal”) did not show an impact on GMDS at 2 and 3 years, while “cross-sectional” EUGR at 6 months seemed to be associated with a worse neurodevelopment for some areas of GMDS at 2 and 3 years, particularly for 2 y locomotor, 2 y personal/social and 2–3 y language (Table 4 and Table 6) development. “Longitudinal” EUGR from TEA to 6 months of CA was associated with a worse GMDS in 2 y performance and 3 y practical reasoning (Table 4 and Table 6). Furthermore, we showed that a better weight z-score at 6 months is protective for developmental delay and borderline global GMDS at 2 years of age (Table 3). The time point at 6 months was not taken into consideration in recent existing studies; in particular, Zozaya et al. compared “longitudinal” and “cross-sectional” EUGR at 36 weeks of CA [8], and De Rose et al. compared 48 definitions of EUGR—24 “cross-sectional” and 24 “longitudinal” at different time points from births to TEA [10]. Furthermore, the above-mentioned studies did not consider MRI data as an influencing factor for the outcome.

A result similar to ours was published by Guellec I et al. as part of the EPIPAGE study. They demonstrated an association between “longitudinal” EUGR from birth to 6 months of age and cerebral palsy at 5 years in patients born appropriate for gestational age AGA and with cognitive deficiency and school difficulties in SGA at 5–8 years, even if not significantly [13]. This study, although extremely cited, also did not consider analyzing lesions with MRI to predict the neurological follow-up for their subjects.

Our results reinforce the focus on not only the EUGR but to post-discharge growth, focusing again on the feeding problems of neonates with VLBW. Indeed, although there are evidence-based recommendations on VLBW nutrition in hospitals [41], little is known about the optimal nutrition instructions after discharge and also the introduction of solid food (referred to as weaning) guidelines.

Regarding post-discharge nutrition, in our center, we recommend breast milk feeding. Exclusive post-discharge breastfeeding is rarely possible in our preterm babies, so the initial indication is to feed them with unfortified breast milk via a bottle for a hydric quotient of 160 up to 200 mL/kg per day. In the absence of breast milk, a post-discharge formula is prescribed for an energy quotient of between 120 and 140 kcal/kg/day until TEA and 3000 g of weight are reached; then, it is replaced with common formula milk for an energy quotient of between 110 and 130 kcal/kg/day. These intakes are adjusted and individualized based on weight growth in the last weeks of hospital stay, the first few weeks post-discharge and depending on the presence of co-morbidities. Unfortunately, breast milk was available for 59% of the neonates, but only 17% were exclusively fed by breast milk at 40 weeks of CA in our population, which is a poor result that is similar to other studies [42]. There is a large consensus that breast milk represents the best choice due to its well-known positive effects on neurodevelopment and body composition [41,43,44,45,46]. The role of breast milk fortification and the choice of better formula milk after discharge is still controversial. As a matter of fact, two quite recent Cochrane meta-analyses did not provide evidence that the fortification of breast milk after hospital discharge or the use of post-discharge formula milk (enriched in protein, LCPUFA, and micronutrients) are able to differently affect the growth rates and neurodevelopmental outcomes of neonates [47,48,49]. Furthermore, with regard to nutrition in the immediate post-discharge period, the only intervention that shows encouraging results is to support and educate families to promote breast milk feeding [42].

Weaning is likely to be crucial for infantile nutrition and neurodevelopment [50,51], although there is great variability in time of introduction, micronutrient supplementation and types of foods proposed from center to center [52]. In our population, weaning started at 6 months of postnatal age (PA), according to the ESPHGAN indications [53]. In the literature, different timing for weaning was proposed from the results of various observational studies [54,55,56]. The few randomized controlled trials (RCTs) available in the literature do not report significant differences in weight growth when weaning is started at 4 vs. 6 months of age [57] or at 13 vs. 17 weeks of age, where an improvement in the body length at 12 months of age is the only result [58]. Only one recent RCT found that starting weaning in VLBW at 10–12 weeks of CA instead of 16–18 weeks of CA had positive effects on the weight z-score at 6 months [59]. None of these RCTs considered neurodevelopment as the primary outcome. The efficacy on the growth of this “early weaning policy” is still controversial and has poor evidence, as observed in different cohort studies [60,61], but it seems to be safe as it does not increase the risk of obesity [61,62] and food allergy or atopic dermatitis [63]. A recent systematic review by the Italian Societies of Pediatrics (SIP), Neonatology (SIN), and Paediatric Gastroenterology, Hepatology and Nutrition (SIGENP) tried to draw up recommendations for weaning in preterm infants and recommended to start weaning between 5 and 8 months of postnatal age and to consider a limit of 3 months of CA to ensure that newborn infants acquire appropriate developmental skills [64]. Based on our data and what is reported in the literature, we speculate that an early weaning policy at about 3 months of age, when developmental skills are acquired (i.e., between about 4 and 6 months of PA depending on GA), will be safe and could improve weight gain at 6 months and the neurological outcomes of patients with VLBW.

Data about the type of food proposed in weaning in preterm infants are lacking, so the guidelines used for term babies are those used for neonates too [53,65]. Interestingly, the Cochrane systematic review reports a decrease in the risk of undernutrition and growth improvements when the family of term babies receive adequate nutritional education, although the effects on neurodevelopment remain uncertain [66]. A similar study on the families of neonates was conducted but it did not provide any convincing evidence, although we can speculate that similar effects can be reached in preterm infants [67]. Proposing adequate nutritional education to families of preterm infants may improve the growth and neurodevelopment of these patients.

Lastly, other well-known risk factors like GA [4,5,6], male sex [68,69], NEC [28,29] and major surgeries [33,34] were identified and confirmed by our study as the risk factors for worse GMDS scores in different areas. Regarding MRI findings, in our population, PWML was confirmed to be a risk factor for worse GMDS scores, particularly for locomotor development (scale A) and performance (scale E) at 2 years, which is in agreement with previous findings [24]. PWML was not significant at 3 years of GA. This result deviates from our previous study due to the differences in the classification of MRI lesions, patient selection and statistical analyses [70].

The strengths of our study comprise the large population of neonates with VLBW up to 3 years of age and the comparisons made between the two groups: one with absolutely no brain lesions (at MRI) and the other with minor lesions whose long-term clinical significance remains understudied.

## 5. Conclusions

Our study confirms that extra-uterine growth restriction affects the neurological outcomes in preterm infants. EUGR diagnosed at 6 months of age seems to have a major negative impact on many areas of neurological development, more than those deriving from minor brain lesions that are almost exclusively diagnosed with MRI. Only periventricular white matter lesions show a significantly negative effect similar to EUGR, although this factor was restricted to locomotor development and performance and, exclusively, at 2 years follow-up. At 3 years of age, EUGR at 6 months maintained a negative effect on language development and practical reasoning. The issue of EUGR and nutrition during NICU stay is well known, and we cannot undervalue its importance on the outcomes of patients, but we believe that it is appropriate to focus more attention on post-discharge nutritional help by optimizing the nutritional education of families.

## Figures and Tables

**Table 1 nutrients-16-00449-t001:** Population details.

Whole Population	*n* = 288
Gestational age (weeks)	28.9 ± 2.1
Birth weight (g)	1097 ± 255 g (z-score −0.449 ± 1.09)
Small for gestational age (SGA)	62 (21.5%)
Male sex	139 (48.3%)
Cesarean delivery	232 (80.6%)
Apgar at 5 min	8 ± 1.2
Sepsis	108 (37.5%)
Necrotizingenterocolitis (NEC)	30 (10.4%)
Bronchodysplasia (BPD)	68 (23.6%)
Major surgery	36 (12.5%)
NEC surgery	15 (5.2%)
Patent ductus arteriosus surgery	17 (5.9%)
Exclusive mother milk feeding	49 (17%)
Exclusive formula feeding	118 (41%)
MRI low-grade lesions	101 (35.1%)
Low-grade intraventricular hemorrhage (GMH-IVH)	44 (15.3%)
Punctate white matter lesions (PWMLs)	47 (16.3%)
Cerebellar micro-hemorrhage (micro-CBH)	31 (10.8%)
Weight at term age TEA (g)	2600 ± 598 (z-score −1.407 ± 1.415)
“Cross-sectional” EUGR at TEA	144 (50%)
“Longitudinal” EUGR at TEA	126 (43.8%)
Weight at 6 months (kg)	6.81 ± 1.03 (z-score −1.240 ± 1.29)
“Cross-sectional” EUGR at 6 months	133 (46.2%)
“Longitudinal” EUGR at 6 months	46 (16.0%)
Weight at 12 months (kg)	8.78 ± 1.16 (z-score −1.360 ± 1.25)
“Cross-sectional” EUGR at 12 months	139 (48.3%)

Data are reported in mean value ± SDS for continuous variables, absolute number and percentage for categorical variables.

**Table 2 nutrients-16-00449-t002:** Griffith Mental Development Scale II (GMDS) results at 2 and 3 years in the whole population for global and sub-scales development quotient (DQ).

**Developmental Quotient < 85 GMDS at 2 y (*n* = 288)**	
Global DQ	56 (19.4%)
Locomotor (scale A)	68 (23.6%)
Personal/social (scale B)	67 (23.2%)
Language (scale C)	127 (44.1%)
Hand–eye coordination (scale D)	38 (13.2%)
Performance (scale E)	70 (24.3%)
**Developmental Quotient < 85 GMDS at 3 y (n = 262)**	
Global DQ	100 (38.2%)
Locomotor (scale A)	83 (31.7%)
Personal/social (scale B)	87 (33.2%)
Language (scale C)	139 (53.0%)
Hand–eye coordination (scale D)	83 (31.6%)
Performance (scale E)	126 (48.1%)
Practical reasoning (scale F)	101 (38.5%)

**Table 3 nutrients-16-00449-t003:** Univariate and multivariate analysis results for global development quotient at 2 years.

2 y GMDS Global DQ	<85	≥85	
*n*	56	232	Total 288
z-score at 6 months	−1.639 ± 1.582	−1.140 ± 1.188	*p* = 0.03
z-score at 12 months	−1.735 ± 1.56	−1.264 ± 1.155	*p* = 0.03
Surgical NEC	6 (10.7%)	9 (3.9%)	*p* = 0.08
“Cross-sectional” EUGR at 6 months	33 (58.9%)	101 (43.5%)	*p* = 0.07
**Multivariate analysis (corrected for GA):**			
**z-score at 6 months**	**OR 0.74 (CI 95% 0.59–0.93)**		***p* = 0.01**

Univariate and multivariate analysis corrected for gestational age (GA); global development quotient (DQ) at 2 years Griffith Mental Development Scale. Data are reported in mean value ± SDS for continuous variables, absolute number and percentage for categorical variables. Odds ratios (OR) are calculated with a 95% confidence interval (CI). All *p*-values are based on two-tailed tests.

**Table 4 nutrients-16-00449-t004:** Multivariate analysis for different Griffith Mental Development Subscales at 2 years.

**2 y GMDS Locomotor DQ**		
Major surgery	OR 3.79 (CI 95% 1.69–8.49)	*p* = 0.001
**“Cross-sectional” EUGR at 6 months**	**OR 1.96 (CI 95% 1.10–3.47)**	***p* = 0.02**
Punctate white matter lesions (PWMLs)	OR 2.33 (CI 95% 1.15–4.71)	*p* = 0.02
**2 y GMDS Personal/social DQ**		
**“Cross-sectional” EUGR at 6 months**	**OR 1.94 (CI 95% 1.12–3.37)**	***p* = 0.02**
NEC	OR 2.60 (CI 95% 1.14–5.92)	*p* = 0.02
**2 y GMDS Language DQ**		
NEC	OR 2.48 (CI 95% 1.07–5.71)	*p* = 0.03
**“Cross-sectional” EUGR at 6 months**	**OR 1.87 (CI 95% 1.05–3.29)**	***p* = 0.02**
Weight z-score at birth	OR 0.31 (CI 95% 0.12–0.81)	*p* = 0.02
**2 y GMDS Hand–eye Coordination DQ**		
NEC	OR 3.98 (CI 95% 1.66–9.55)	*p* = 0.002
**2 y GMDS Performance DQ**		
Male sex	OR 2.01 (CI 95% 1.13–3.57)	*p* = 0.02
Punctate white matter lesions (PWMLs)	OR 2.03 (CI 95% 1.00–4.14)	*p* = 0.05
Major surgery	OR 4.07 (CI 95% 1.78–9.33)	*p* = 0.001
**“Longitudinal” EUGR at 6 months**	**OR 2.10 (CI 95% 1.03–4.30)**	***p* = 0.04**

Multivariate analysis corrected for gestational age (GA) for different Griffith Mental Development Subscales at 2 years. Odds ratios (OR) are calculated with a 95% confidence interval (CI). All *p*-values are based on two-tailed tests.

**Table 5 nutrients-16-00449-t005:** Univariate and multivariate analysis results for global development quotient at 3 years.

3 y GMDS Global DQ	<85	≥85	
*n*	100	162	Total 262
Male sex	60 (60%)	70 (43.2%)	*p* = 0.01
Caesarean delivery	76 (76%)	138 (85.2%)	*p* = 0.05
NEC	17 (17%)	12 (7.4%)	*p* = 0.02
**Multivariate analysis (corrected for GA):**			
Male sex	OR 1.94 (CI 95% 1.16–3.24)		*p* = 0.01
NEC	OR 2.55 (CI 95% 1.11–5.86)		*p* = 0.03

Univariate and multivariate analysis corrected for gestational age (GA); global development quotient (DQ) at 3 years Griffith Mental Development Scale. Data are reported in mean value ± SDS for continuous variables, absolute number and percentage for categorical variables. Odds ratios (OR) are calculated with a 95% confidence interval (CI). All *p*-values are based on two-tailed tests.

**Table 6 nutrients-16-00449-t006:** Multivariate analysis for different Griffith Mental Development Subscales at 3 years.

**3 y GMDS Locomotor DQ**		
Male sex	OR 1.82 (CI 95% 1.07–3.10)	*p* = 0.03
**3 y GMDS Personal/social DQ**		
Male sex	OR 2.18 (CI 95% 1.28–3.72)	*p* = 0.004
Caesarean delivery	OR 0.47 (CI 95% 0.25–0.91)	*p* = 0.02
**3 y GMDS Language DQ**		
**“Cross-sectional” EUGR at 6 months**	**OR 1.63 (CI 95% 0.99–2.68)**	***p* = 0.05**
Male sex	OR 1.88 (CI 95% 1.14–3.10)	*p* = 0.01
**3 y GMDS Hand–eye Coordination DQ**		
NEC	OR 4.17 (CI 95% 1.78–9.76)	*p* = 0.001
Male sex	OR 1.80 (CI 95% 1.04–3.10)	*p* = 0.03
**3 y GMDS Performance DQ**		
NEC	OR 4.31 (CI 95% 1.63–11.35)	*p* = 0.003
Male sex	OR 2.39 (CI 95% 1.44–3.97)	*p* = 0.001
**3 y GMDS Practical Reasoning DQ**		
NEC	OR 4.47 (CI 95% 1.84–10.85)	*p* = 0.001
**“Longitudinal” EUGR at 6 months**	**OR 2.07 (CI 95% 1.02–4.17)**	***p* = 0.04**

Multivariate analysis corrected for gestational age (GA) for different Griffith Mental Development Subscales at 3 years. Odds ratios (OR) are calculated with a 95% confidence interval (CI). All *p*-values are based on two-tailed tests.

## Data Availability

Original data are available upon request. The data are not publicly available due to internal privacy policy.

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
