# Peer review of "Early Extra-Uterine Growth Restriction in Very-Low-Birth-Weight Neonates with Normal or Mildly Abnormal Brain MRI: Effects on a 2–3-Year Neurodevelopmental Outcome"

_nutrients, 2024, doi:10.3390/nu16030449_

Round 1

Reviewer 1 Report

Comments and Suggestions for Authors

The aim of the paper has been to assess the impact of extrauterine growth restriction on neurodevelopmental outcome at 2 and 3, in conjunction with term MRI findings in a group of babies born < 35 weeks. The title says VLBW when the group is preterm < 35 weeks. 

However the greatest change in z score was during the NICU stay and I don't think this was adequately addressed in the paper.

The GA is wider than other cohorts and includes babies32 weeks or above.

The follow up after exclusions missed 25.5%. A comparison of this group compared to the included 288 for the neonatal characteristics reported should be done to prove the study children are representative.

Line 110 we collected on weight ….. you have already stated in line 109 that weight was done ….   Do you mean length?

 Line 120 remove the word pathologic - it is not required.

 Line 135 is stated ….. values below 70 define developmental delay. Then in Line 186 that it is “pathologic” or borderline. Be consistent and suggest don't use pathologic. Also used in line 192.

In the results and Table 1, 2, 3, 4,5,6.:  Decimal points have mostly been written as comma’s. Please use decimal point eg 5.9 not 5,9

The discussion focusses on the EUGR. The majority of the growth restriction has occurred between birth and TEA- proportion < 10th% goes from 21.2% at birth to 50% at TEA. Has the analysis compared the impact of birth to TEA change in z score on the Developmental outcomes at 2 and 3? 

I think you are saying that from TEA to 6 months that the longitudinal EUGR rate was 16%, (line 178) compared to birth to TEA longitudinal rate was 43.8%. (Line 176). Please check. 

 In the section Line 213 to 230 a format was used up to line 224 giving the area of result first eg Considering the locomotor are ……. Then for eye hand co-ordination and performance area it was reversed. Please be consistent.

In Table 4 2year language DQ and  In table 6: Multivariate analysis at 3 years  Locomotor: Should GA be included when it was corrected for in the analysis?  

 I’m not sure that the TEA to 6 month z score change is the issue or the birth to TEA growth change.

The significant variates - male sex, NEC, major surgery are as other longitudinal studies have found.

That the punctate WM lesions were no longer significant at 3 years  was not discussed. Confirmation that they are associated with locomotor development was supported by reference 27 and 75 have some of the same authors as this paper. All but one of the 9 studies in the de Bruijn review (27) were assessed at 2 years or less. The conclusions are affected by the studies descriptions and limited numbers. Paper 75 reported on the same cohorts outcomes at 3 years. 

In the conclusion (line 352) remove retardation (an out dated term and not used in your definition in Line 21) change to restriction.

Author Response

The aim of the paper has been to assess the impact of extrauterine growth restriction on neurodevelopmental outcome at 2 and 3, in conjunction with term MRI findings in a group of babies born < 35 weeks. The title says VLBW when the group is preterm < 35 weeks. 

Thank you for the comment. Our population include only preterm babies with a birth weight <1500g, so all the patients include in our population are very low birth weight preterms even if there is a wide range for GA.

However the greatest change in z score was during the NICU stay and I don't think this was adequately addressed in the paper.

Thanks for the comment. Although there is a high incidence of poor growth during NICU hospitalization in our population (43.8%) this seems to have less impact on neurological outcome than growth at 6 months of corrected age, so we focused on the latter. We add a comment of this on discussion: “In our population we found a high incidence of EUGR in particular at TEA (Table 1). Despite this, EUGR at TEA (both “cross-sectional” and “longitudinal”) didn’t show an impact on GMDS at 2 and 3 years while “Cross-sectional” EUGR at 6 months seemed to be associated with a worse neurodevelopment for some areas of GMDS at 2 and 3 years in particular: 2y locomotor, 2y personal-social, 2-3y language (Table 4, 6). “Longitudinal” EUGR from TEA to 6 months of CA was associated with a worse GMDS in 2y performance and 3y practical reasoning (Table 4, 6). Furthermore, we showed that better weight z-score at 6 months is protective for developmental delay-border line global GMDS at 2 years of age (Table 3).”

The GA is wider than other cohorts and includes babies 32 weeks or above.

That’s a good point. We selected the population on the basis of birth weight (VLBW). Within the population there are only a few patients with EG greater than 32 weeks (only 5 patients, complete data avaiable on request) and this can be seen in the text from the mean and SD of the EG (28,9 + 2,1 weeks) (Table 1)

The follow up after exclusions missed 25.5%. A comparison of this group compared to the included 288 for the neonatal characteristics reported should be done to prove the study children are representative.

Thank you for this comment. The populations had comparable EG and birth weight. We add a comment in the manuscript: “498 very low birth weights underwent brain MRIs from January 2012 to December 2018 (mean GA 28,7 + 2,3; mean birth weight 1081g + 265g).”

Line 110 we collected on weight ….. you have already stated in line 109 that weight was done ….   Do you mean length?

Thank you for the comment. No we didn’t consider length in our study. We mean that weight was measured many times during NICU stay and follow up visits but we only collected weight at birth, TEA, 6 months and 12 months for the data analisys in this study. We modify the manuscript according to your specification: “During the hospital stay, weight was measured daily and after discharging it was measured at term-equivalent age (TEA), and 1, 3, 6, 9, and 12 months of correct age. We collected only weight at birth, at TEA, 6 and 12 months of corrected age for the statistical analysis.”

 Line 120 remove the word pathologic - it is not required.

Thank you for the comment. The manuscript has been edited according to your indications: “A weight with z-score <1.282 (<10° percentile) was also defined as EUGR at 12 months of correct age”

Line 135 is stated ….. values below 70 define developmental delay. Then in Line 186 that it is “pathologic” or borderline. Be consistent and suggest don't use pathologic. Also used in line 192.

Thank you for the comment. The manuscript has been edited according to your specifications (Underlined parts in Results)

In the results and Table 1, 2, 3, 4,5,6.:  Decimal points have mostly been written as comma’s. Please use decimal point eg 5.9 not 5,9

Thank you for the comment. The manuscript has been edited according to your indications.

The discussion focusses on the EUGR. The majority of the growth restriction has occurred between birth and TEA- proportion < 10th% goes from 21.2% at birth to 50% at TEA. Has the analysis compared the impact of birth to TEA change in z score on the Developmental outcomes at 2 and 3? 

Thank you for the comment. Yes, we compared the impact of "longitudinal" EUGR (loss of 1 point of SD) from birth to TEA on GMDS at 2 and 3 y and it wasn't significant. In table we reported all the significative results of multivariate analysis. We add a comment on discussion:” Despite this, EUGR at TEA (both “cross-sectional” and “longitudinal”) didn’t show an impact on ….”

.I think you are saying that from TEA to 6 months that the longitudinal EUGR rate was 16%, (line 178) compared to birth to TEA longitudinal rate was 43.8%. (Line 176). Please check. 

Thank you for the comment. We would saying that the incidence of patients that lost 1 SD point from TEA to 6 months was 16%. We modify the manuscript to clarify this point: “The incidence of “longitudinal” birth to TEA was 43.8% (n=126). At 6 months of age, the mean weight was 6.81+ 1.03 kg, z-score -1.24+1.29, the incidence of patients with “cross-sectional” EUGR was 46.2% (n=133), while the rate of "longitudinal" EUGR from TEA to 6 moths”

 In the section Line 213 to 230 a format was used up to line 224 giving the area of result first eg Considering the locomotor are ……. Then for eye hand co-ordination and performance area it was reversed. Please be consistent.

Thank you for your comment. We modified the manuscript in according to your specifications. (Underlined parts in Results)

In Table 4 2year language DQ and  In table 6: Multivariate analysis at 3 years  Locomotor: Should GA be included when it was corrected for in the analysis?

Thank you for your comment. The multivariate analysis were corrected for GA. In according with your comment we remove  this data in from table 4 and 6 and text.

I’m not sure that the TEA to 6 month z score change is the issue or the birth to TEA growth change.

Thank you for the comment. In our population we considered 3 time point of EUGR diagnosis, TEA, 6 months and 12 months. The multivariate analysis showed that only the 6 months time point ( cross sectional and longitudinal) have an impact on GMDS. We think that EUGR in general have an impact on GMDS but our data shown that the 6 months time point may have major significance. We modified conclusions in according to your comment: The issue of EUGR and nutrition during NICU stay is well known and we can’t undervalue its importance on outcome, but we believe that it’s appropriate to focus more attention on post-discharge nutritional help, optimizing family nutritional education.”

The significant variates - male sex, NEC, major surgery are as other longitudinal studies have found. That the punctate WM lesions were no longer significant at 3 years was not discussed. Confirmation that they are associated with locomotor development was supported by reference 27 and 75 have some of the same authors as this paper. All but one of the 9 studies in the de Bruijn review (27) were assessed at 2 years or less. The conclusions are affected by the studies descriptions and limited numbers. Paper 75 reported on the same cohorts outcomes at 3 years. 

Thank you for your comment. The aim of our study is not to focus on the brain lesions, in fact we selected a population without major lesions. In this paper we exclude only the patients with cystic PVL while in the previous paper (Malova et al) there was a differentiation between mild- moderate and severe WML so the result at 3 years is different because there is a different analysis and population. We add a comment of this in discussion: PWML was not significant at 3 years of GA. This result deviates from our previous study due to differences in MRI lesions classification, patient selection and statistical analysis [70]”

In the conclusion (line 352) remove retardation (an out dated term and not used in your definition in Line 21) change to restriction.

Thank you for the comment. We modify conclusions in according to your indications: Our study confirms that extrauterine growth restriction affects neurological outcomes in preterm infants.”

Reviewer 2 Report

Comments and Suggestions for Authors

Dear authors,

This is a very interesting study evaluating Extrauterine growth restriction as a common complication and a known risk factor for impaired development in very low birth weight preterms. You demonstrated that EUGR at 6 months was associated with worse locomotor, personal-social, language and performance DQ at 2 years and worse language and practical reasoning DQ at 3 years. Your suggestion is a better investment on post-discharge  nutritional, optimizing family nutritional education. I think your work is important and deserves to be considered after a major revision. I have a few suggestions: 

-You want your work to be published in Nutrients, and in the Conclusions chapter you draw attention to the role of nutrition. In this case, in the introduction chapter, I think the role of nutrition in the neuropsychomotor development of VLBW children should be mentioned and discussed.

Methods: - the statistics paragraph deserves a subheading

Results: - what was the cause of such a high cesarean delivery rate (80,6%)?

References 14 and 38 are the same. Also please reduce the rate of self-citation.

Author Response

This is a very interesting study evaluating Extrauterine growth restriction as a common complication and a known risk factor for impaired development in very low birth weight preterms. You demonstrated that EUGR at 6 months was associated with worse locomotor, personal-social, language and performance DQ at 2 years and worse language and practical reasoning DQ at 3 years. Your suggestion is a better investment on post-discharge  nutritional, optimizing family nutritional education. I think your work is important and deserves to be considered after a major revision. I have a few suggestions: 

-You want your work to be published in Nutrients, and in the Conclusions chapter you draw attention to the role of nutrition. In this case, in the introduction chapter, I think the role of nutrition in the neuropsychomotor development of VLBW children should be mentioned and discussed.

Thank you for the comment. We add a brief discussion of the role of nutrition in introduction in according with your indication: “Nutritional strategy adopted in the NICU can play an important role not only in the prevention of EUGR but also on the neurological development of VLBW patients. In particular, the use of fortified breast milk or, as second choice, enriched formulas for premature infants for enteral nutrition and the early start of parenteral nutrition appear to have a positive impact on preterms neurodevelopment. However, there is lack of evidence regarding the role of post-discharge nutrition [16].”

Methods: - the statistics paragraph deserves a subheading

Thank you for your comment. We edited the manuscript in according to your specifications.

Results: - what was the cause of such a high cesarean delivery rate (80,6%)?

That’s a good observation. The incidence its quite higher than some other report in literature (Plevani C, Incerti M, Del Sorbo D, Pintucci A, Vergani P, Merlino L, Locatelli A. Cesarean delivery rates and obstetric culture - an Italian register-based study. Acta Obstet Gynecol Scand. 2017 Mar) but there similar incidence in some preterm population (Karayel Eroglu H, Gulasi S, Mert MK, Cekinmez EK. Relationship between the mode of delivery, morbidity and mortality in preterm infants. J Trop Pediatr. 2022 Oct 6;68(6):fmac074.)  We don’t know why there’s a so high incidence of cesarean in our population (maybe it was linked with prematurity with mother-fetal complications?) but  we think that this data cannot influence our results.

References 14 and 38 are the same. Also please reduce the rate of self-citation.

Thank you for your comment. We modified the manuscript in according to your indications and the self-citation rate now is decrease from 22,8% to 13%. Excluded references:

  • Sannia A, Natalizia AR, Parodi A, Malova M, Fumagalli M, Rossi A et al. Different gestational 270 ages and changing vulnerability of the premature brain. J Matern Fetal Neonatal Med 2015; 28 271 Suppl 1: 2268–72.
  • Parodi A, Giordano I, De Angelis L, Malova M, Calevo MG, Preiti D, Ravegnani M, Cama A, Bellini C, Ramenghi LA. Post-haemorrhagic hydrocephalus management: Delayed neonatal transport negatively affects outcome. Acta Paediatr. 2021 Jan;110(1):168-170. .
  • Malova M, Morelli E, Cardiello V, Tortora D, Severino M, Calevo MG, Parodi A, De Angelis LC, Minghetti D, Rossi A, Ramenghi LA. Nosological Differences in the Nature of Punctate White Matter Lesions in Preterm Infants. Front Neurol. 2021 Apr 29;12:657461. doi: 10.3389/fneur.2021.657461. PMID: 33995255; PMCID: PMC8117674.

Round 2

Reviewer 1 Report

Comments and Suggestions for Authors

The authors have addressed the issues raised

Reviewer 2 Report

Comments and Suggestions for Authors

Dear authors,

Thank you for your answers. Your manuscript is improved and after some minor typographical corrections it can be published.